# Fast Filling of Microvia by Pre-Settling Particles and Following Cu Electroplating

**DOI:** 10.3390/nano12101699

**Published:** 2022-05-16

**Authors:** Ganglong Li, Zhiyi Li, Junjie Li, Houya Wu

**Affiliations:** 1School of Mechanical and Electronic Engineering, East China University of Technology, Nanchang 330013, China; liganglong@ecut.edu.cn; 2State Key Laboratory of High Performance Complex Manufacturing, School of Mechanical and Electrical Engineering, Central South University, Changsha 410083, China; lizhiyi@csamq.com; 3Shenzhen Institute of Advanced Electronic Materials, Shenzhen Institute of Advanced Technology, Chinese Academy of Sciences, Shenzhen 518055, China

**Keywords:** microvia filling, nanoparticle, electroplating, advanced packaging

## Abstract

Microvia interconnectors are a critical element of 3D packaging technology, as they provide the shortest interconnection path between stacked chips. However, low efficiency of microvia filling is a long-standing problem. This study proposed a two-step method to enhance the electroplating filling efficiency by pre-setting metal particles in microvias and later electroplating the Cu to fill the gaps among the pre-settled particles. Since these particles occupy a certain volume in the microvia, less electroplating Cu is needed for microvia filling, leading to a shorter electroplating period.

## 1. Introduction

As the integrated circuit (IC) devices keep scaling down, the three-dimensional (3D) packaging has become the inevitable trend in the development of IC device packaging technology [1]. A microvia interconnector is a critical element in the stacking packaging process, as it provides the shortest interconnection path between stacked chips [2]. The blind microvia is usually made by deep reactive ion etching on the silicon substrate. Then, the filling of the microvia is completed by two steps of nanometer-thick Cu seed layer deposition and Cu electroplating [3]. However, low efficiency is the major issue in the Cu electroplating process [4,5]. Many studies have investigated methods to enhance the filling rate of the microvia, such as adjusting concentration of the electrolyte solution [6], higher current density input [7] or introducing ultrasound stirring [8], but filling defects are possibly induced.

This study proposed an innovative method to fill the microvia by firstly pre-setting Ag materials and the following Cu electroplating to enhance the Cu-filling efficiency. Similar to Cu, Ag is also an interconnection material with high conductivity and high thermal conductivity. Cu/Ag composites are also often used in semiconductor packaging interconnection [9], printed circuits [10] and microvia filling [11]. Therefore, Ag is used as the pre-filling material in the experimental design of this paper, which meets the requirements of interconnect material performance in the microelectronic packaging process. Through this two-step method, the filling speed of microvia has been greatly improved; the filled structure is integrated and almost without any defects, which is expected to improve the packaging efficiency in practical application.

## 2. Materials and Methods

The chip containing microvia and Cu seed layer was purchased from National Center for Advanced Packaging Co., Ltd., Wuxi, China, raw materials of electrolyte were from Jiangsu Mengde New Materials Technology Co., Ltd., Danyang, China, and the Ag particles were from Beijing Deke Daojin Science and Technology Co., Ltd., Beijing, China.

Microvia is filled by two steps: step 1, particles are first deposited in the microvia; step 2, Cu is electroplated to fill the gaps among the particles in the microvia, where a composite of Cu and particles is formed (Figure 1). 

Process of step 1: a chip (10 mm × 20 mm) containing blinding microvias (20 μm × 60 μm, opening upward) is firstly immersed in a beaker containing the suspension of ethanol and particles; the beaker is then placed in a vacuum device for 3 min to evacuate bubbles from the microvia and facilitate the particles transporting into the deep region of the microvia; after that, the beaker is moved to an ultrasound vibration device for 3 min to further release the residual bubbles and enhance particle settling; the vacuum and the ultrasound vibration treatment is repeated three times; finally, the chip is taken out, and the particles landing on the surface of the chip are removed by a plastic wiper. It is worth noting that an additional sintering process can be used if necessary.

Process of step 2: Cu is electrodeposited to fill the gaps among the particles pre-settled in the microvia by step 1. Four groups of experiments are conducted to investigate the factors affecting the filling process of the microvia, including the microvia filling method, particle size, and sintering temperature. Each experiment is repeated three times. The electrolyte composition and working condition of the electroplating process is shown in Table 1.

## 3. Results and Discussion

In this paper, we chose two sizes of Ag particles, 20 nm and 600 nm, for pre-filling and study. According to the size effect, the smaller the nanoparticles, the easier it is to achieve sintering to form porous structures. In this work, the small-size 20 nm Ag particles were used in this experiment, which can form porous structures with characteristic sizes of 100–500 nm after sintering at different temperatures. At the same time, other 600 nm Ag nanoparticles for direct pre-filling without sintering were also selected for comparison with the sintered structure of small Ag nanoparticles. However, when the size of Ag particles increases further, they are difficult to be prefilled into the microvias by the ultrasonic process. Therefore, we did not use larger Ag particles for comparison.

For convenience of presentation, we define the sample without pre-settled Ag particles as sample #1; the sample with pre-settled 600 nm Ag particles as sample #2; the sample with pre-settled 20 nm Ag particles as sample #3; and samples with sintered 20 nm Ag nanoparticles at different sintering temperatures of 200 °C, 300 °C and 400 °C as sample #4-1, #4-1 and #4-1, respectively. The microvia of sample #1 is fully filled by using the traditional electroplating method within 80 min (Figure 2a), where no particle is pre-settled in the microvia. 

### 3.1. Effects of Particle Size

When Ag particles of 600 nm are pre-settled in the microvia, composites of Cu and Ag particles are obtained, which are evenly distributed in the microvia (Figure 2b). However, when the Ag particle size is 20 nm, an inhomogeneous distribution of the particles appears in the microvia. These Ag particles are mainly concentrated near the bottom, sidewalls, and the microvia opening, leaving no Ag particle in the centre region of the microvia. It is possible that the small Ag particles are still movable in the microvia during the electroplating process. As the electroplating Cu grows, the small Ag particles are excluded from their original locations by the newly electroplated Cu. However, when these Ag particles are close to the boundaries, they have less freedom and can be buried inside the Cu layer near the boundary of the microvia. In addition, the burying of small Ag particles in the opening region is possibly caused by the slow Cu-electroplating rate at this region (a high concentration of PEG leads to a slow electroplating rate of Cu at the opening). By contrast, the large Ag particles are less movable due to gravity and cannot be excluded away from their location. Therefore, to obtain uniformly distributed particles in the microvia, the mobility of the particles should be limited.

### 3.2. Effects of Sintering Temperature

To limit the movement of the small particles in the microvia, we added a sintering process after the particle-wiping process. The samples were sintered for 10 min at 200 °C, 300 °C, and 400 °C, respectively. After sintering, Ag particles (raw material 20 nm) can be fixed in the microvia as they form a porous network structure (Figure 2d). 

The microvias were then filled by Cu electroplating. Although the Ag particles were evenly distributed in the microvia after 200 °C sintering, the quantity of Cu filled in the microvia was very limited and mainly existed in the opening region (Figure 2e). A similar result was obtained as the sintering temperature increased to 300 °C (Figure 2f). However, when the sintering temperature increased to 400 °C, the microvia was successfully filled by Cu from the opening to the bottom, and the Ag particles were evenly doped in the Cu layer (Figure 2g). 

The porous structures of the sintered Ag particles with different sintering temperatures are observed, as shown in Figure 3a–c. When the sintering temperature was 200 °C, the grain sizes of the sintered powder were fine, and the gaps among the grains were small. However, as the sintering temperature rose, the grain sizes of the sintered powder became larger. When the sintering temperature was 300 °C, parts of the particles became large, and a porous network structure started to form. At 400 °C, almost all of the Ag particles had been fused together, and a porous network structure was formed.

The porosity and pore size of the porous structure of the Ag particles are the major factors influencing the Cu ions transporting in the microvia, while the porosity and pore size are affected by the sintering temperature. We processed the images of Figure 3a–c to determine the porosity of the porous structures, as correspondingly shown in Figure 3d–f. At 200 °C, 300 °C and 400 °C, the estimated porosity is 25.7%, 28.0% and 30.2%, respectively.

According to Tertre [12] and Gao [13], higher porosity and larger pore size denote higher permeability, facilitating the transportation of the electrolyte mass through the porous structures. At 200 °C and 300 °C, due to the low porosity and small pore size of the porous structure, the Cu ions are unable to be transferred into the deep region of the microvia, and the electrodeposited Cu layer is mainly in the opening region. At 400 °C, the porosity of the porous structure is higher, and the Cu is evenly distributed throughout the microvia. It is worth noting that higher sintering temperatures than 400 °C have not been investigated, as the factory annealing temperature of the microvia interconnect is typically around 420 °C [14].

### 3.3. Filling Rate Comparison

Figure 4a shows that a time of 80 min is needed to fill the microvia using the traditional method. However, with particles pre-settled in the microvia, the electroplating periods are only 35 min, 70 min, and 40 min for samples #2, #3, and #4-3, respectively. The total time combining particle pre-settling (white column) and electroplating (colorful column) of sample #2 is only 53 min.

With particles pre-settled in the microvia, only gaps among the particles need to be filled by electroplating Cu, which directly shortens the electroplating time for microvia filling. Figure 4c shows a reverse correlation between the electroplating time and the volume percentage of the particles. It only takes only 35 min of electroplating to fill the microvia when 56.9 Vol% Ag particles are pre-settled in the microvia (sample #2).

## 4. Conclusions

In this paper, a two-step process was developed to enhance the microvia-filling efficiency. The pre-settled Ag structures achieved by filling large Ag particles or the sintering of small Ag nanoparticles can significantly shorten the electroplating time for microvia filling. With this method, an integrated filled microvia structure with Cu/Ag composites can be obtained, which presents fewer defects and shows strong application prospects.

## Figures and Tables

**Figure 1 nanomaterials-12-01699-f001:**
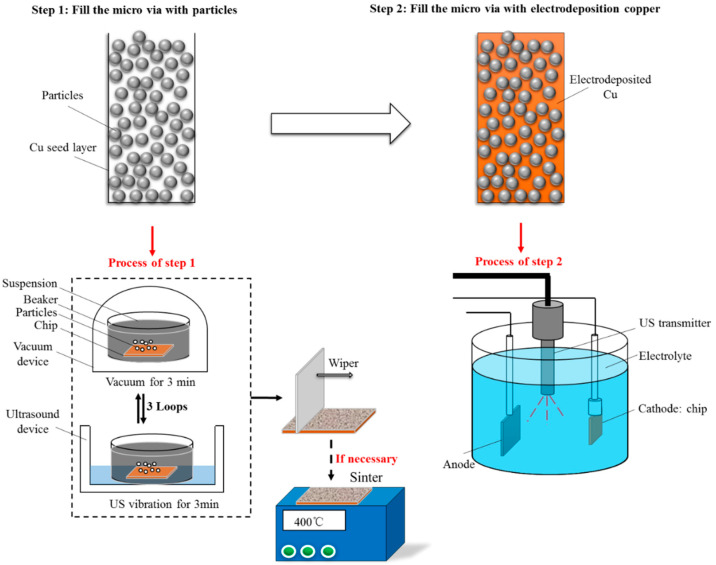
Scheme of the two-step method of microvia filling.

**Figure 2 nanomaterials-12-01699-f002:**
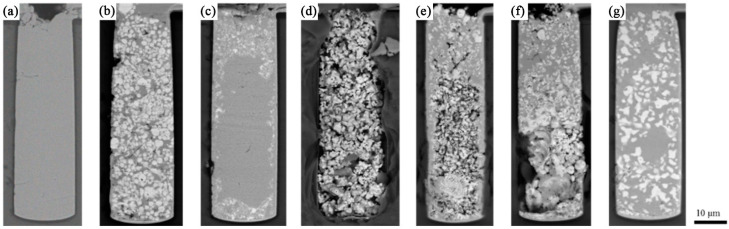
Cross-section of microvias: (**a**) sample #1, (**b**) sample #2, (**c**) sample #3, (**d**) particles sintered in microvia, (**e**) sample #4-1, (**f**) sample #4-2, (**g**) sample #4-3.

**Figure 3 nanomaterials-12-01699-f003:**
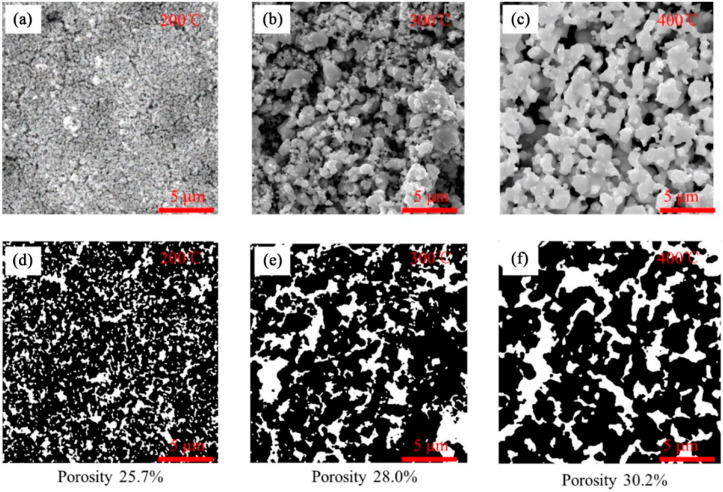
Sintered Ag powder: (**a**) 200 °C, (**b**) 300 °C, (**c**) 400 °C; (**d**–**f**) are images of (**a**–**c**) processed using Photoshop CS6, respectively.

**Figure 4 nanomaterials-12-01699-f004:**
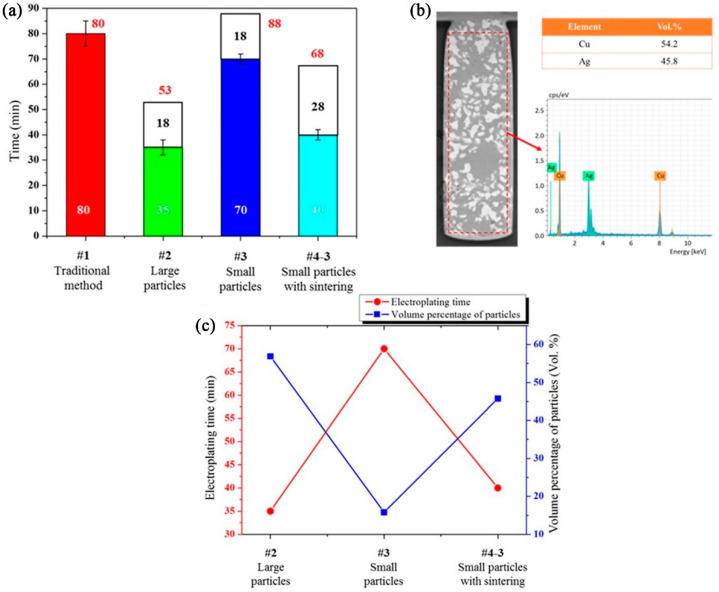
(**a**) Filling time of microvias using different methods, (**b**) particle content detected by EDS, (**c**) relationship between electroplating time and particle contents.

**Table 1 nanomaterials-12-01699-t001:** Electrolyte composition and working conditions.

Electrolyte Composition	
CuSO_4_·5H_2_O	195 g/L
NaCl	0.1 g/L
H_2_SO_4_	32 mL/L
PEG	0.3 g/L
SPS	0.1 g/L
PNI	0.1 g/L
**Electroplating Working Condition**	
Current density	0.2 A/dm^2^
Ultrasound (20 kHz)	90 W

## Data Availability

The data presented in this study are available on request from the corresponding author. The data are not publicly available due to the original data may involve some technical secrets.

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
