# Peer review of "Fast Filling of Microvia by Pre-Settling Particles and Following Cu Electroplating"

_nanomaterials, 2022, doi:10.3390/nano12101699_

Round 1

Reviewer 1 Report

The Communication with title “Fast Filling of Microvia by Pre-Settling Particles and Following Cu Electroplating” presents interesting results.

The paper is well written and well detailed.

The English is good.

However, there are some minor comments that should be taken into consideration:

  1. The Introduction part, is too short with only five references regarding others works. The newest reference from this work is from 2020. But, according to Web of Science or Scopus search, there are at least 6 new papers published between 2021-2022 regarding the “microvia filling”.

Please mentioned why your results are better than those obtained by other authors.

  1. In the Conclusions part, the authors affirmed that “The pre-settled large particles (raw material or sintered) can effectively shorten the electroplating time for microvia filling” but they didn’t use large particles with sintering. Only small particles were combined with the sintering process.

Please correct this error.

Author Response

Response to Reviewer 1:

Comment:

The introduction part, is too short with only five references regarding others works. The newest reference from this work is from 2020. But, according to Web of Science or Scopus search, there are at least 6 new papers published between 2021-2022 regarding the “microvia filling”.

Please mentioned why your results are better than those obtained by other authors.

Response:

Thanks very much for your comments and suggestions. We have already added more reference papers that published in recent years. The introduction has been extended and the advantages of our process have been highlighted, as described in line 32 to 41. The added reference papers were highlighted from line 177 to 199.

Comment:

In the Conclusions part, the authors affirmed that “The pre-settled large particles (raw material or sintered) can effectively shorten the electroplating time for microvia filling” but they didn’t use large particles with sintering. Only small particles were combined with the sintering process.

Please correct this error.

Response:

Thanks for your suggestions. Our description in the original manuscript does tend to cause ambiguity. Therefore, we have changed the expression to " The pre-settled Ag structures achieved by filling large Ag particles or sintering of small Ag nanoparticles can significantly shorten the electroplating time for microvia filling." as shown in line 160 to 161.

Reviewer 2 Report

Review report

Manuscript title: Fast Filling of Microvia by Pre-Settling Particles and Following Cu Electroplating

Manuscript ID: nanomaterials-1708024

Review comments: The work is good and interesting. The work demonstrates a two-step method to enhance the electroplating filling efficiency by pre-setting metal particles in microvias and later electroplating the Cu to fill the  gaps among the pre-settled particles. Since these particles occupy a certain volume in the microvia, less electroplating Cu is needed for microvia filling, leading to a shorter electroplating period. I think it will be published in the current journal after modification the issues raised. 

  1. In the Introduction part needs to be more explanation and comparison to others research work. It is too short. Need more references added.
  2. In table 1, the authors use different concentration of electrolyte solution. Why is it not same concentration?
  3. Figure 4 should be rearranged, upper two figures are good, and the downpart can be in the middle.
  4. The English needs to be rechecked and make it more corrections. 

Author Response

Response to Reviewer 2:

Comment:

In the Introduction part needs to be more explanation and comparison to others research work. It is too short. Need more references added.

Response:

Thanks for your suggestion. The introduction has been extended and the advantages of our process have been highlighted, as described in line 32 to 41. The added reference papers were highlighted from line 177 to 199.

Comment:

In table 1, the authors use different concentration of electrolyte solution. Why is it not same concentration?

Response:

Thanks for your comment. I think the different concentration you are referring to means that the concentration units of various components are different, right? The main reason for this difference is that several components in the electrolyte include powder and liquid reagents, and it is more convenient to use different units for weighing to ensure process consistency.

Comment:

Figure 4 should be rearranged, upper two figures are good, and the downpart can be in the middle. The English needs to be rechecked and make it more corrections.

Response:

Thanks for your carefully reading and kindly suggestion. We have rearranged Figure 4 and also revised some inappropriate language expressions, as highlighted in the revised manuscript.

Reviewer 3 Report

  1. The introduction is clearly too short and doesn’t describe sufficiently a state-of-the art. It must be extended.
  2. To pre-set particle you used a suspension of ethanol and Ag particles. Did you consider to use another material for the particles?
  3. You used 20 and 600 nm particle. Did you use particle of diameter that is between 20 and 600 nm. If yes, what the results are? If no, why you did not use them?
  4. In microvias you obtained composites of Cu and Ag. Correct?

What is a typical concentration Cu and Ag within the composite?

How stable the composite is?

  1. Does the obtained composite is compatible with further processing, post-processing?
  2. What was the Cu seed layer?
  3. The Table 1. Electrolyte Composition and working conditions. It is difficult to read. It must be split, at least in 3-4, such as following: Electrolyte Composition, Particles description etc.Is it any reason to have one table?

Author Response

Response to Reviewer 3:

Comment:

The introduction is clearly too short and doesn’t describe sufficiently a state-of-the art. It must be extended.

Response:

Thanks for your suggestion. The introduction has been extended and the advantages of our process have been highlighted, as described in line 32 to 41. The added reference papers were highlighted from line 177 to 199.

Comment:

To pre-set particle you used a suspension of ethanol and Ag particles. Did you consider to use another material for the particles?

Response:

Thanks for your comment. Yes, we have considered other materials. However, Ag material is more suitable for this study, because Ag is also a metal material with very good thermal conductivity as Cu, and its microstructure can be controlled by the sintering process, as the added description in line 33 to 38

Comment:

You used 20 and 600 nm particle. Did you use particle of diameter that is between 20 and 600 nm. If yes, what the results are? If no, why you did not use them? In microvias you obtained composites of Cu and Ag. Correct? What is a typical concentration Cu and Ag within the composite?

How stable the composite is?

Response:

Thanks for your comments. We have added some description for the selecting of Ag materials, as described in the paragraphs added in the manuscript, “in this paper, we have chosen two sizes of Ag particles, 20 nm and 600 nm, for pre-filling and study. According to the size effect, the smaller the nanoparticles, the easier it is to achieve sintering to form porous structures. In this work, the small size 20 nm Ag particles are used in this experiment, which can form porous structures with characteris-tic sizes of 100-500 nm after sintering at different temperatures. At the same time, another 600 nm Ag nanoparticles for direct pre-filling without sintering were also selected for comparison with the sintered structure of small Ag nanoparticles. However, when the size of Ag particles increases further, they are difficult to be prefilled into the microvias by the ultrasonic process. Therefore, we did not use larger Ag particles for comparison.”

For Cu/Ag composite, it is always used as a stable material in semiconductor packaging interconnect, printed electronics and microvia filling, as described in the highlighted sentences in introduction part.

Comment:

Does the obtained composite is compatible with further processing, post-processing? What was the Cu seed layer?

Response:

In general, whether the microvia filling process can be matched with the subsequent CMP process mainly depends on whether the filling structure has defects. The filling integrity in this work is very good, and in principle it can match the subsequent process. However, specific research will be carried out in the follow-up cooperation with packaging factories.

In addition, the Cu seed layer is the substrate used to induce Cu deposition on the walls of the microvias. The manufacturing process of this Cu seed layer is already completed by the company when the chip is purchased, as described in the added sentence in line 43.

Comment:

The Table 1. Electrolyte Composition and working conditions. It is difficult to read. It must be split, at least in 3-4, such as following: Electrolyte Composition, Particles description etc. Is it any reason to have one table?

Response:

Thanks for your kindly suggestions. We split a large table into a small table along with some descriptive sentences for a clearer understanding, as shown in line 43-46, line 71-83.

Round 2

Reviewer 3 Report

No further comments.